# Longitudinal Evaluation of DCE-MRI as an Early Indicator of Progression after Standard Therapy in Glioblastoma

**DOI:** 10.3390/cancers16101839

**Published:** 2024-05-11

**Authors:** Julio Arevalo-Perez, Andy Trang, Elena Yllera-Contreras, Onur Yildirim, Atin Saha, Robert Young, John Lyo, Kyung K. Peck, Andrei I. Holodny

**Affiliations:** 1Department of Radiology, Memorial Sloan Kettering Cancer Center, 1275 York Ave, New York, NY 10065, USA; atrang@ucla.edu (A.T.); elenyllera@gmail.com (E.Y.-C.); yildirio@mskcc.org (O.Y.); sahaa@mskcc.org (A.S.); youngr@mskcc.org (R.Y.); peckk@mskcc.org (K.K.P.); holodnya@mskcc.org (A.I.H.); 2Department of Radiology, Weill Medical College of Cornell University, 525 East 68th Street, New York, NY 10065, USA; 3Brain Tumor Center, Memorial Sloan Kettering Cancer Center, 1275 York Ave, New York, NY 10065, USA; 4Department of Medical Physics, Memorial Sloan Kettering Cancer Center, 1275 York Ave, New York, NY 10065, USA; 5Department of Neuroscience, Weill-Cornell Graduate School of the Medical Sciences, 1300 York Ave, New York, NY 10065, USA

**Keywords:** dynamic contrast-enhanced (DCE) perfusion MRI, maximum plasma volume Vp max, Ktrans, GBM, progression of disease, POD

## Abstract

**Simple Summary:**

Our study explores the capability of dynamic contrast-enhanced MRI (DCE-MRI) as an early predictor of disease progression in glioblastoma patients, compared to conventional MRI. We analyzed two cohorts comprising 26 patients with newly diagnosed primary glioblastoma. Patients were then categorized based on disease stage (progression or stability) and underwent three DCE-MRI scans prior to progression or consecutively for stable disease. Parameters such as Ktrans and plasma volume (Vp) were measured within the volume of interest (VOIs). Our findings revealed a gradual increase in Vp max values preceding disease progression in patients who underwent routine MRI scans. Utilizing quantitative DCE-MRI may offer an opportunity to detect disease progression earlier than conventional methods, potentially improving patient management by enabling proactive measures against progression.

**Abstract:**

**Background and Purpose:** Distinguishing treatment-induced imaging changes from progressive disease has important implications for avoiding inappropriate discontinuation of a treatment. Our goal in this study is to evaluate the utility of dynamic contrast-enhanced (DCE) perfusion MRI as a biomarker for the early detection of progression. We hypothesize that DCE-MRI may have the potential as an early predictor for the progression of disease in GBM patients when compared to the current standard of conventional MRI. **Methods:** We identified 26 patients from 2011 to 2023 with newly diagnosed primary glioblastoma by histopathology and gross or subtotal resection of the tumor. Then, we classified them into two groups: patients with progression of disease (POD) confirmed by pathology or change in chemotherapy and patients with stable disease without evidence of progression or need for therapy change. Finally, at least three DCE-MRI scans were performed prior to POD for the progression cohort, and three consecutive DCE-MRI scans were performed for those with stable disease. The volume of interest (VOI) was delineated by a neuroradiologist to measure the maximum values for Ktrans and plasma volume (Vp). A Friedman test was conducted to evaluate the statistical significance of the parameter changes between scans. **Results:** The mean interval between subsequent scans was 57.94 days, with POD-1 representing the first scan prior to POD and POD-3 representing the third scan. The normalized maximum Vp values for POD-3, POD-2, and POD-1 are 1.40, 1.86, and 3.24, respectively (FS = 18.00, *p* = 0.0001). It demonstrates that Vp max values are progressively increasing in the three scans prior to POD when measured by routine MRI scans. The normalized maximum Ktrans values for POD-1, POD-2, and POD-3 are 0.51, 0.09, and 0.51, respectively (FS = 1.13, *p* < 0.57). **Conclusions:** Our analysis of the longitudinal scans leading up to POD significantly correlated with increasing plasma volume (Vp). A longitudinal study for tumor perfusion change demonstrated that DCE perfusion could be utilized as an early predictor of tumor progression.

## 1. Introduction

Glioblastoma multiforme (GBM) is the most common and aggressive type of primary brain tumor in adults. The outcome of GBM patients remains poor, with a dismal survival of 14 months, despite modern treatments [1]. Young age, absence of neurological signs, complete surgical resection, and good performance status are favorable prognostic factors [2]. The current standard of care consists of maximal safe surgical resection followed by radiotherapy with concomitant and adjuvant temozolomide [1,2].

Imaging of GBMs during treatment requires an understanding of the tumor microenvironment, where angiogenesis plays a fundamental role. Traditionally, the standard for diagnosis of recurrence of these tumors is the change in contrast-enhancement on brain magnetic resonance imaging (MRI) [3] and is largely guided by the response assessment in neuro-oncology (RANO) criteria [4]. However, these size-based criteria are limited since enhancement may simply represent post-treatment changes with disruption of the blood–brain barrier and increased vessel leakiness instead of neovascularity and angiogenesis characteristic of growing tumors.

Dynamic contrast enhancement (DCE) MR imaging, as an emerging advanced MR technique, offers a noninvasive characterization of the tumor vascular microenvironment and hemodynamics (ref) that is not available from conventional MR imaging and the traditional dynamic susceptibility contrast (DSC or T2*) perfusion method. Such information from DCE-MRI has been utilized as a biomarker to differentiate high vs. low-grade tumors, differentiate recurrent GBM from radiation necrosis and pseudoprogression, differentiate between various brain metastases, and evaluate the treatment effects of anti-angiogenic therapy and radiation therapy [5,6,7,8,9,10,11,12,13,14].

The purpose of our study is to evaluate the utility of DCE-MRI as a biomarker for the early detection of progression in patients with GBM while the routine scans remain unchanged. We hypothesize that DCE-MRI can play an important role as an early predictor for the progression of disease in GBM patients when compared to the current standard of conventional MRI utilized by radiologists and neuro-oncologists.

## 2. Methods

### 2.1. Patient Demographics

This retrospective study was authorized by the institutional review board and granted a Waiver of Informed Consent. In full compliance with all Health Insurance Portability and Accountability Act regulations, we identified patients diagnosed with glioblastoma who received standard therapy (maximal resection, RT/TMZ, and adjuvant TMZ) from 2011 to 2023. We identified 26 patients with proven glioblastoma by histopathology and gross or subtotal resection of the tumor. Then, we classified them into 2 groups: 16 patients with clinical POD confirmed by pathology or change in chemotherapy and 10 patients with stable disease without evidence of progression or need of therapy change. Finally, we identified at least three DCE-MRI scans over 6 months prior to POD for the progression cohort and three consecutive DCE-MRI scans for those with stable disease. Of note, the 3 scans obtained before progression had no evidence for progression on routine conventional MRI.

### 2.2. Scan Timing and Nomenclature

Scans are indicated as progression of disease minus 1 scan (POD-1) for the 1st scan prior to progression of disease, progression of disease minus 2 scans (POD-2) for the 2nd scan prior, and progression of disease minus 3 scans (POD-3) for the 3rd scan prior (therefore, POD-3 is the earliest scan). The mean interval between subsequent scans was 57.94 days (SD = 27.99).

### 2.3. MRI Acquisition

MRI sequences were acquired as a part of a standard clinical protocol with a GE Premier MR 750w 3T scanner (GE Healthcare, Milwaukee, WI, USA) and a standard 8-channel head coil. DCE-MRI of the brain was acquired upon completion of routine MRI scans. A power injector was used to administer a bolus of Gadobutrol (Bayer, Barmen, Germany) administered at 0.1 mmol/kg body weight and a rate of 2–3 mL/s, with 40 cc saline. The kinetic enhancement of tissue before, during, and after injection of gadolinium-diethylenetriamine penta-acetic acid was obtained using a 3-dimensional T1-weighted fast spoiled-gradient echo sequence [repetition time (TR), 4–5 ms; echo time (TE), 1–2 ms; slice thickness, 3 mm; no slice gap; flip angle (FA), 25°; FOV, 24 cm; matrix, 256 × 256; temporal resolution of 5~6 s; phase encoding direction A-P; bandwidth 35.71; parallel acceleration with phase (×1.5) and slice (×1.25)] and consisted of 400 to 2074 images in the axial plane. Ten phases for pre-injection time delay and thirty phases for post-injection were obtained. DCE matching post-contrast T1-weighted spin-echo images (TR/TE = 600/8 ms; thickness = 4.5 mm) were obtained after DCE-MRI

### 2.4. Image Analysis

Data processing and analysis were performed using dynamic image processing software (version 2.3.14; NordicNeuroLab, Bergen, Norway). Preprocessing steps included the removal of background noise, spatial and temporal smoothing, and automatic detection of the arterial input function (AIF) from the aorta. Spatial smoothing using 4 mm isotropic Gaussian kennel and temporal smoothing using a low pass filter was used to reduce image noise and spikes in dynamic signal response. AIF was individually calculated in each acquisition of every patient. The appropriate shape of the AIF curve was visually confirmed before the processing steps continued. Pixels with a large change in signal intensity, a rapid change immediately after bolus injection, and an early peak in intensity were chosen for AIF. The linear assumption was made between the change in signal intensity and gadolinium concentration to convert the signal intensity curve to the concentration–time curve. DCE-matching T1-weighted spin-echo images that matched the DCE-MR images were used for region of interest (ROI) analysis.

ROIs (regions of interest) were manually delineated by a single trained operator (medical student), supervised by a neuroradiologist with 15 years of experience, including the entire region of tumor enhancement on the post-contrast T-1 weighted images. The operators were blinded to the results. Since, according to the literature, maximal perfusion values are the most precise for tumor grading, ROIs were then transferred to Vp and Ktrans parametric maps for every slice of the tumor, creating a volume of interest (VOI) [11,15,16,17,18,19,20]. Based on Toft’s extended two-compartment pharmacokinetic model, VOIs were used to obtain the Vp maximum value (Vp max) and the Ktrans maximum value (Ktrans max) and to assess inter-scan changes. For normalization purposes, a ratio of tumor to normal cerebral parenchyma was obtained for both Vp max and Ktrans max by placing ROIs in healthy white/gray matter of the contralateral hemisphere over normal brain parenchyma.

### 2.5. Statistical Analysis

A non-parametric Friedman test of differences among repeated measures was conducted between the parametric perfusion maps at the three different time points to obtain a Friedman Statistic (FS). The percent change (and the respective standard error) in Vp and Ktrans were calculated between each scan.

## 3. Results

### 3.1. Patient Population

Twenty-six patients (age range 44–79, IQR 16, mean 61.2, and SD 10.03) with the diagnosis of glioblastoma and DCE-MRI perfusion scans were identified. They were classified according to the progression of the disease: 16 patients (9 male [56.25%] and 7 women [43.75%]); 10 patients (8 men [80%]) and 2 women [20%]) with stable disease.

### 3.2. Conventional MRI Findings

Conventional MRI and particularly post gadolinium T1-weighted sequences (T1+C) demonstrated no significant interval change in size or appearance of the lesions in the analyzed patients, which was read as a stable disease in all three scans prior to the clinical diagnosis of progression of disease, POD-3, POD-2, and POD-1 (Figure 1).

### 3.3. Quantitative Perfusion Analysis

Vp values correlated with an increase towards progression of disease for maximum Vp. The normalized maximum Vp values leading up to progression for POD-3, POD-2, and POD-1 were 1.40, 1.86, and 3.24, respectively (FS = 18.00, *p* = 0.0001) (Figure 2).

Time-dependent leakage (Ktrans) measurements demonstrated an increasing trend prior to POD between T-1 and T-2 time points; however, the opposite was observed between T-2 and T-3. Despite the trends, there were no significant differences between the measurements for Ktrans maximum values (FS = 1.13, *p* < 0.57) (Figure 2).

The stable cohort demonstrated no statistically significant increase between T-1 and T-2 time points or between T-2 and T-3 time points (Figure 2).

## 4. Discussion

The results of this study demonstrate that Vp max values progressively increase in the three scans prior to POD when measured on routine MRI scans (FS = 18.00, *p* = 0.0001).

Histologic findings of recurrent GBM are characterized by vascular proliferation and neoangiogenesis. On imaging, enhancing lesions of recurrent GBM corresponds to a combination of neoangiogenesis, vascular proliferation, and neoplastic blood–brain barrier (BBB) disruption with increased permeability to macromolecular contrast agents. Accurate diagnosis is paramount in order to make informed treatment decisions. A diagnosis of POD indicates that the current treatment is ineffective and should prompt a change in therapy, such as repeat surgery and/or new chemotherapy [7].

Prior studies have addressed the search for an early biomarker of progression with advanced MRI techniques. Khalifa et al. found that the ratio of hypoperfused volume to the total tumor volume in GBM treated with chemotherapy and radiation therapy could be considered a relevant biomarker that could predict progression at the following examination with a specificity of 63.6% and a sensitivity of 92.3% [21]. They hypothesized that the reduction in hypoperfused volume fraction could be more sensitive and happen earlier than the increase in hyperperfused volume, thus anticipating progression of disease. They also added, in agreement with other studies, the idea that hypoxia could be a sanctuary for cancer cells responsible for tumor growth and relapse [22,23]. In their study, with a comparable number of patients (n = 25), they used DSC-perfusion, an advanced MRI technique that relies on rCBV to assess indirectly the vascularity of brain tumors. DSC perfusion imaging is more sensitive to susceptibility artifacts and can be easily affected by hemorrhage and calcifications, common findings in treated gliomas that could affect the estimation of both hyper and hypoperfused volumes. There is also a potential bias in measurements because of the T1 effects from extravascular contrast leakage in GBM neovasculature. From a technical perspective, it is acknowledged that rCBV values derived from DSC are a semi-quantitative measurement that can be influenced by various post-processing steps, including correction techniques to address contrast extravasation and the choice of normal contralateral white matter.

In our study, we used DCE-MRI, a more robust and stable noninvasive perfusion technique that is less prone to artifacts and provides quantitative pharmacokinetic parameters reflecting the microcirculatory physiology of the tumor. The most used parameters are Vp, considered an imaging biomarker for angiogenesis, and K_trans_, which reflects permeability [24]. We evaluated the use of DCE-MRI as an early indicator for progression of disease for patients with GBM who have undergone resection and standard therapy. We found that the increasing Vp measurements before progression were non-linear, as there was a significantly greater increase in Vp max between POD-1 and POD-2 time points (91.70% increase, SE = 28.07), in comparison to POD-2 and POD-3 time points (43.08% increase, SE = 15.11). This greater increase in Vp between POD-1 and POD-2 is congruent with what we would expect, as the tumor is likely to become more vascular, with increased expression of VEGF as it approaches progression and is accompanied by neo-angiogenesis [25]. Time-dependent leakage (Ktrans) measurements demonstrated an increasing trend prior to POD between POD-1 and POD-2 time points, but this was not statistically significant. These results for the Ktrans measurements may be a consequence of the chemotherapy and radiation treatment-related effects that would cause variation in vascular permeability and leakage rather than reflect the true anatomy [26,27]. Another possible explanation would be based on the fact that the BBB, in a physiologic setting, has a shielding effect on neurons by serving as a highly selective permeability barrier, but it is also a key obstacle to the successful delivery of therapeutic agents to brain tumors [24]. Ktrans is a surrogate for permeability or BBB integrity, except in flow-limited conditions when Ktrans represents the blood plasma flow per unit volume of tissue instead of vascular permeability [28]. Yoo et al. hypothesized that an enhancing lesion with decreased Ktrans values was more likely to progress because it might represent the part with low permeability or leakiness, where the delivery of TMZ to the viable tumor cells might have been less successful during standard treatment [24].

### Limitations of the Study

First, our study was conducted retrospectively with the possibility of selection bias. We required three scans prior to progression in order to document potential changes in perfusion over time. While necessary for longitudinal analysis, this excludes patients who progressed very rapidly, requiring a change in treatment; therefore, our results may not apply in a rapidly progressing cohort. Second, the post hoc image analysis was performed on a limited subset of patients. Despite the limited sample size, it is the first of its kind to longitudinally compare via serial scanning quantitative data from DCE-MRI in GBM. Third, VOIs were manually delineated, and although this represents the standard of care for perfusion analysis, it is an operator-dependent technique with inevitable potential interobserver and intra-observer variability. However, by having a single trained operator delineate the ROIs, the potential for variation was mitigated. Despite these limitations, we observed significant results that make future prospective studies worthwhile. On another note, one of the most important factors affecting the estimation of kinetic model parameters in DCE perfusion is how to convert MRI signal intensities to contrast concentration. As a limitation in this study, linearity between signal and contrast concentration was assumed. However, in DCE perfusion MRI, the linear relationship between signal intensity and contrast concentration is validated only for a small concentration because of the T1 shortening effect. A future study including non-linear conversion based on MRI signals in terms of accuracy should be followed.

## 5. Conclusions

Quantitative DCE-MRI may be a useful biomarker for early detection of progression of disease in glioblastomas. The potential of quantitative DCE-MRI to detect progression of disease earlier than the current standard may make it a worthwhile metric to consider in patient management, allowing clinicians to take earlier proactive measures against progression. Our results may be hypothesis-generating for a further study to assess the importance and utility of early assessment of perfusion parameters.

## Figures and Tables

**Figure 1 cancers-16-01839-f001:**
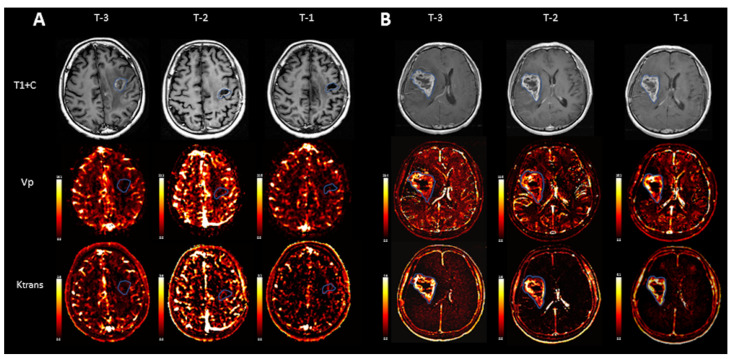
Representative scans from two individual patients, a patient with stable disease (**A**) and a patient with progression of disease (**B**). While T1 weighted images after contrast (T1+C) demonstrate no gross significant interval change in size or appearance of the tumor over time in a patient with POD (**B**), quantitative evaluation of Vp demonstrates an increment in normalized Vp max values between the first and last scans before POD: 2.63 (T-3), 2.43 (T-2), and 3.97 (T-1). Normalized Ktrans max also demonstrated an increase between the first and last scans before progression: 3.49 (T-3), 1.43 (T-2), and 4.24 (T-1).

**Figure 2 cancers-16-01839-f002:**
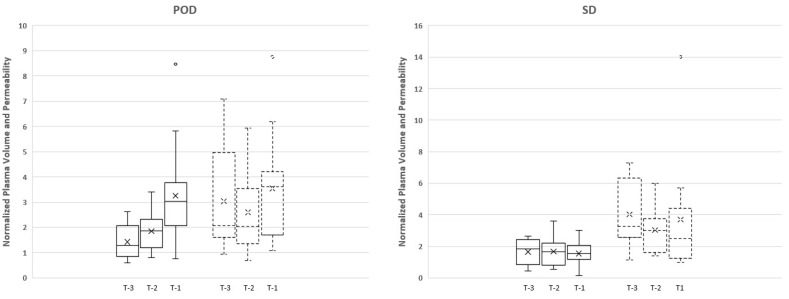
The boxplot demonstrates the data distribution of normalized plasma volume (solid line) and permeability (dotted line) in three time points (T-3, T-2, and T-1) for each group (progression (POD) and stable (SD) group). There was a greater increase in Vp maximum between T-1 and T-2 time points (74% increase) in comparison to T-2 and T-3 time points (33.3% increase). Note that the *x*-axis (T-3, T-2, T-1) is in chronological order from older to recent scans prior to progression; O: outlier; X: mean.

## Data Availability

Data are contained within the article.

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
