# Peer review of "Longitudinal Evaluation of DCE-MRI as an Early Indicator of Progression after Standard Therapy in Glioblastoma"

_cancers, 2024, doi:10.3390/cancers16101839_

Round 1
Reviewer 1 Report
Comments and Suggestions for Authors
The manuscript is well described however in my opinion I think the authors need to look into the references which describes the DCE-MRI rCBV and rCBF as a potential parameters for grading brain tumors. Also, only vp and ktrans results are described and their is no supporting evidence of any other MRI parameter like ADC, SWI and T2W-Flair.
Comments:
Introduction:
Please cite the references (Gupta et al. Neuroradiology 2021), (Sengupta et. al, JMRI 2019) that represents the potential of rCBV and rCBF in grading and differentiating tumors.
line 56, page 2 ref is missing
Methods:
How were the sample size calculated(power analysis). There is no evidence of that.
why were the T2W and FLAIR images not acquired that is a standard in clinical setting to evaluate edema
Image analysis should atleast be done by 2 operators to evaluate the intra observer variability. Were the trained operator blinded.
Why was the SWI not acquired to see any obvious haemorrhage or calcification
There are reference available that use semi-automated delineation of tumor.
Results
There are no color bar available for the Ktrans and Vp maps.
The results evaluated here are just with TMZ. No evidence how many dose were given to the patient. what was the first Vp and Ktrans value when the patient was diagnosed with GBM and how it has changed when disease progressed with treatment same with the stable patient.
Analysis can be separated based on age because the their is a big difference in the upper ad lower age limit.
The author has to come up with more supporting data.
Author Response
The response to Reviewer 1 has been attached.

Reviewer 2 Report
Comments and Suggestions for Authors
Quantitative dynamic contrast enhancement MRI (DCE-MRI) may be a useful biomarker for early detection of progression of disease in glioblastomas. In this study the analysis of the longitudinal scans using this technique leading up to progression of disease, significantly correlated with increasing plasma volume (Vp). Therefore, DCE perfusion could be utilized as an early predictor of tumor progression. The potential of quantitative DCE-MRI to detect progression of disease earlier than the current standard may make it a worthwhile metric to consider in patient management, allowing clinicians to take earlier proactive measures against disease progression. However, the importance and utility of early assessment of perfusion parameters in management of patients with these tumors remains unclear. These tumors routinely progress, and the treatment options are limited, and therefore earlier knowledge about the progression of the disease will unlikely result in improvement in patient outcome.
Comments on the Quality of English LanguageMinor issues only.
Author Response
Thank you for highlighting this aspect. We acknowledge your perspective on this matter. This preliminary study is designed to underline some of the potential benefits offered by the DCE-MRI sequence in GBM monitoring. Glioblastoma multiforme (GBM) indeed stands as the most prevalent and aggressive primary intracranial tumor. Despite advancements in therapy, its prognosis remains bleak, with a median survival of only 14 months. While mean survival and progression-free survival (PFS) are commonly used as primary metrics of response, it is imperative to evaluate the impact of treatments on disease burden and the quality of life of patients. Alongside the evolution of therapeutic protocols, advancements in diagnostic imaging have played a pivotal role in managing GBM. During follow-up, an increasing number of GBM patients undergoing treatment have encountered radiologic findings suggestive of disease progression (pseudoprogression), introducing a new challenge in tumor management. Distinguishing pseudoprogression from true progression is crucial due to the substantially different management strategies they may require. In cases of pseudoprogression, continued follow-up is warranted, with the patient considered stable. Conversely, true progression necessitates treatment adjustments. Enhanced radiological techniques and advancements in discerning pseudoprogression from true progression are paramount for optimizing both quality of life and prognosis. In this regard, Dynamic contrast-enhanced magnetic resonance imaging (DCE-MRI) shows promise in differentiating between pseudoprogression and true progression. The ability to identify distinct disease-free intervals warrants further investigation into subsequent lines of treatment. Although we may not be able to alter the outcome of the illness itself, we still play a crucial role in assessing tumor management, monitoring follow-up, and framing disease-free intervals. This aids clinicians in exploring new treatments and ultimately extending patients' lives.
Reviewer 3 Report
Comments and Suggestions for Authors
Abstract
During the time of the study the WHO definition of glioblastoma changed. How did you include this in the study design ? Please state how many patient were included with which definition.
Page 1 Line 23:
“Then we classified them in 2 groups; patients with progression of disease (0) confirmed by pathol-“
«(0)» should be «(POD)»
Methods
The two patients groups were not matched test if any other factor could explain the difference in Ktrans for T-3
2.3. MRI Acquisition
Why was the B1 correction not included in the MRI acquisition?
Discuss potential effects and the literature on this.
Give more of the sequence parameters: BW, parallel acceleration method and factor, water-fat shift in voxel, slice spacing or gap, phase encoding direction, etc.
How much Saline solution was used the flush the Gd contrast agent and at which rate?
2.4 Image analysis
Provide all details on the noise canceling, and temporal and spatial smoothing: filter name and function and values of the parameters.
Discuss the assumption of linearity between signal and contrast agent concentration. This is only true for a very small concentration range.
How was the supervision of the medical student drawing the ROIs performed? Were all ROIs corrected/controlled?
Please provide a second rater and rater correlation analysis.
Figure 2
Provide the histogram representation showing all subjects values and the box-whiskers graph.
Author Response
The response to Reviewer 4 has been added. Please find it attached below.

Round 2
Reviewer 1 Report
Comments and Suggestions for Authors
The authors have edited the manuscript as suggested. The only comment is what about the GBM IDH-wild type and IDH-mutant. Did the authors tried to find any change in Ktrans and Vp values between these two.
Author Response
The authors have edited the manuscript as suggested. The only comment is what about the GBM IDH-wild type and IDH-mutant. Did the authors tried to find any change in Ktrans and Vp values between these two.
Response 1:
Thank you for pointing this out. We concur with the comment raised. All patients were diagnosed, between 2011 and 2022 and all 26 patients included in our study were classified according to the most up to date WHO classification for tumors of the CNS available at the time of diagnosis. This means that some patients were classified according to 2016 but others were classified according to older versions of the classification, like the 2007 WHO CNS tumors classification, that did not distinguish between IDH-wild type and IDH-mutant and only included histological features and were simply classified as glioblastoma (grade VI). Consequently, we cannot classify our patients according to mutant or wild type, and therefore we cannot investigate changes in Ktrans and Vp values between these two.